# Characterization of Co-Formulated High-Concentration Broadly Neutralizing Anti-HIV-1 Monoclonal Antibodies for Subcutaneous Administration

**DOI:** 10.3390/antib9030036

**Published:** 2020-07-29

**Authors:** Vaneet K. Sharma, Bijay Misra, Kevin T. McManus, Sreenivas Avula, Kaliappanadar Nellaiappan, Marina Caskey, Jill Horowitz, Michel C. Nussenzweig, Michael S. Seaman, Indu Javeri, Antu K. Dey

**Affiliations:** 1IAVI, 125 Broad Street, New York, NY 10004, USA; VSharma@iavi.org (V.K.S.); SAvula@iavi.org (S.A.); 2CuriRx, Inc., 205 Lowell Street, Wilmington, MA 01887, USA; bjmisra@gmail.com (B.M.); nellaiappan@curirx.com (K.N.); Ijaveri@curirx.com (I.J.); 3Center for Virology and Vaccine Research, Beth Israel Deaconess Medical Center, Boston, MA 02215, USA; kmcmanu4@bidmc.harvard.edu (K.T.M.); mseaman@bidmc.harvard.edu (M.S.S.); 4Laboratory of Molecular Immunology, The Rockefeller University, New York, NY 10065, USA; mcaskey@mail.rockefeller.edu (M.C.); jhorowitz@mail.rockefeller.edu (J.H.); nussen@mail.rockefeller.edu (M.C.N.); 5Howard Hughes Medical Institute, The Rockefeller University, New York, NY 10065, USA

**Keywords:** HIV/AIDS, co-formulation, high concentration, analytical characterization, antibody (s)

## Abstract

The discovery of numerous potent and broad neutralizing antibodies (bNAbs) against Human Immunodeficiency Virus type 1 (HIV-1) envelope glycoprotein has invigorated the potential of using them as an effective preventative and therapeutic agent. The majority of the anti-HIV-1 antibodies, currently under clinical investigation, are formulated singly for intra-venous (IV) infusion. However, due to the high degree of genetic variability in the case of HIV-1, a single broad neutralizing antibody will likely not be sufficient to protect against the broad range of viral isolates. To that end, delivery of two or more co-formulated bnAbs against HIV-1 in a single subcutaneous (SC) injection is highly desired. We, therefore, co-formulated two anti-HIV bnAbs, 3BNC117-LS and 10-1074-LS, to a total concentration of 150 mg/mL for SC administration and analyzed them using a panel of analytical techniques. Chromatographic based methods, such as RP-HPLC, CEX-HPLC, SEC-HPLC, were developed to ensure separation and detection of each antibody in the co-formulated sample. In addition, we used a panel of diverse pseudoviruses to detect the functionality of individual antibodies in the co-formulation. We also used these methods to test the stability of the co-formulated antibodies and believe that such an approach can support future efforts towards the formulation and characterization of multiple high-concentration antibodies for SC delivery.

## 1. Introduction

The number of approved monoclonal antibodies (mAbs) for therapy against various cardiovascular, cancer, respiratory, hematology, and autoimmune diseases is continuously on the rise [1]. In addition to therapy against non-infectious diseases, monoclonal antibodies are also increasingly seen as potent prophylactic and therapeutic agents against several infectious pathogens [2,3,4,5], particularly those against which effective vaccines do not exist or are under arduous development. To date, over a hundred antibodies have been approved by various regulatory authorities; the majority of these antibody products are typically administered by intravenous (IV) infusion. IV administration, although a well-established route, is challenging to patients as well as to healthcare professionals. Subcutaneous (SC) administration, on the other hand, is increasingly becoming a clear patient preference due to time savings and potential for self-administration, including possibilities for healthcare professionals of administrating during home visits to patients [6,7]. 

The use of monoclonal antibodies as prophylactic and therapeutic options is particularly attractive against Human Immunodeficiency Virus type 1 (HIV-1) [8,9], a viral pathogen for which the development timeline for a prophylactic vaccine is uncertain [10,11,12]. Therefore, protection using passive administration of broadly neutralizing antibodies (bNAbs) against HIV-1 is being evaluated through multiple human clinical studies to test the validity of the approach. Broadly neutralizing (monoclonal) antibodies (bNAbs) such as VRC01 [13,14], 10-1074 [15]/10-1074-LS [16], 3BNC117 [17]/3BNC117-LS [18], VRC07-523-LS [19], PGT121 [20,21], and PGDM1400 [21] or their combinations are currently under investigation in multiple clinical trials. Recent studies by Bar-On et al. [22] and Mendoza et al. [23] showed that the combination of two bNAbs, 3BNC117 (directed to CD4-binding site epitope on HIV-1 surface envelope glycoprotein) [17,24] and 10-1074 (directed to V3-glycan epitope on HIV-1 surface envelope glycoprotein) [15,24], delivered by the intravenous (IV) route was well-tolerated and effective in maintaining virus suppression for extended periods in individuals harboring HIV-1 strains sensitive to the antibodies. These clinical studies, with safety, pharmacokinetics and viral load re-bound or decay as endpoints, have primarily used antibodies formulated for IV infusion. Moving forward, to overcome the high cost and burden of intra-venous administration, the high-concentration formulation of both antibodies (here referred to as co-formulation) for sub-cutaneous (SC) administration is planned. However, co-formulating two (or more) antibodies at high concentration is not only challenging due to the requirement to maintain their optimal quality attributes, low viscosity and stability in the chosen formulation condition but also in developing analytical methods that allow separation of individual antibodies to characterize their quality attributes and measure their individual and total stability [25]. Recently, Cao et al. reported the characterization of antibody charge variants and the development of ”release” assays for co-formulated antibodies [26]. In another study, Patel et al. investigated the formulation of two anti-HIV bNAbs and through a series of analytical tools, including the mass spectrometry-based multi attribute method (MAM), the authors highlight the analytical challenges in the characterization of co-formulated antibodies [27]. 

Here, we describe the formulation of two high-concentration bnAbs, 3BNC117-LS and 10-1074-LS, to a final concentration of 150 mg/mL and characterize them through a panel of analytical methods to evaluate the suitability of the methods for future cGMP testing of the co-formulated drug product. Additionally, we show that the chromatography-based separation methods (RP-HPLC, SE-HPLC and IEX-HPLC) and virus-based neutralization assay are optimal to study each antibody in the co-formulated milieu and can potentially be used for “release” and ”stability” testing of these materials.

## 2. Materials and Methods 

### 2.1. Materials

#### 2.1.1. Monoclonal Antibodies

3BNC117 is a monoclonal antibody of the IgG1κ isotype that specifically binds to the CD4 binding site (CD4bs) within HIV-1 envelope gp120. The bnAbs 10-1074 is of the IgG1λ isotype that specifically targets the V3 glycan supersite within HIV-1 envelope gp120. Both fully human parental monoclonal antibodies, 10-1074 and 3BNC117, were LS-modified, two amino acid substitutions, Methionine (M) to Leucine (L) at Fc position 428 (M428L) and Asparagine (N) to Serine (S) at Fc position 434 (N434S), to enhance the antibody binding affinity to the neonatal Fc receptor (FcRn) and prolong their half-life in mammals without impacting the antibody binding domain or its interaction with antigens [28,29]. The LS-modified monoclonal antibodies (MAbs) are referred to here as 3BNC117-LS and 10-1074-LS. The 3BNC117-LS and 10-1074-LS mAbs were produced at Celldex Therapeutics (Fall River, MA, USA). Both antibodies were expressed via stable Chinese hamster ovary (CHO) cell line clones in a serum-free medium in a batch bioreactor using standard mammalian cell culture techniques. The harvested clarified supernatant was then used to purify the mAbs using a series of chromatographic steps that included MabSelect Sure, Sartobind Q, and SP Sepharose cation exchange column chromatography’s. The SP Sepharose eluate was nano-filtered using Virosart HG filtration and concentrated to 150 mg/mL concentration by UFDF (Ultra-filtration Dia-filtration).

The 3BNC-117-LS monoclonal antibody concentrated to 150 mg/mL was formulated in a buffer containing 10 mM Methionine, 250 mM Trehalose, 0.05% Polysorbate 20, pH 5.2. The 10-1074-LS monoclonal antibody concentrated to 150 mg/mL was formulated in a buffer containing 5 mM Histidine, 250 mM Trehalose, 10 mM Methionine, 5 mM Sodium Acetate, 0.05% Polysorbate 20, pH 5.5.

For this study, 3BNC117-LS and 10-1074-LS, were co-formulated (1:1) by mixing at ambient conditions, and the buffer was exchanged such that the final formulation buffer was 5 mM Histidine, 250 mM Trehalose, 10 mM Methionine, 5 mM Sodium Acetate, 0.05% Polysorbate 20, pH 5.5.

#### 2.1.2. Reagents

The hybridoma-based monoclonal anti-idiotype antibodies, used in the ELISA, were produced at Duke Human Vaccine Institute (Durham, NC, USA). The hybridomas were created by immunizing BALB/C mice with either 10-1074 Fab fragment or 3BNC117 Fab fragment. The generated anti-idiotype antibodies were chromatographically purified and concentrated to ~7 mg/mL in 1 × PBS pH 7.2, 0.22 μm filtered and stored at 4 °C until further use. The USP grade Histidine, Methionine, Polysorbate 20 were purchased from JT Baker Chemicals (Phillipsburg, NJ, USA) and Trehalose was purchased from Pfanstiehl, Inc. (Waukegan, IL, USA). All solutions were stored at 4 °C until used.

### 2.2. Methods

#### 2.2.1. Reverse Phase High-Performance Liquid Chromatography (RP-HPLC)

RP-HPLC separation was performed on Agilent 1260 Infinity Quaternary LC coupled to a diode array detector (DAD). Best peak resolution was demonstrated using Agilent AdvanceBio RP-mAb Diphenyl, 2.1 × 100 mm column, 0.5 mL/min flow rate, with a column temperature of 60 °C and a step wise gradient (3 min washing at 35% B followed by 35% B to 39% B over 16 min). The eluted peaks were detected at 280 nm.

#### 2.2.2. Ion Exchange (IEX)—HPLC

IEX-HPLC was performed on the Agilent 1260 Infinity Quaternary LC system equipped with a diode array detector (DAD) and coupled to ProPac WCX-10, 250 × 4 mm column (Thermo Scientific, Sunnyvale, CA, USA) maintained at 30 °C. Mobile phase A consisted of 20 mM Acetate, pH 5.2, while mobile phase B was 20 mM Acetate, 300 mM sodium chloride, pH 5.2. The pHs of both mobile phases was adjusted using 0.1 M NaOH solution. The flow rate was 0.7 mL/min and salt gradient separation, 50% to 100% B in 35 min, was performed. Peak detection was carried out at 280 nm and the peaks were integrated and percentage peak areas of each peak (as well as charge variants i.e., acidic/basic species) calculated corresponding to each mAbs.

#### 2.2.3. Size-Exclusion High-Performance Liquid Chromatography (SE-HPLC)

SE-HPLC was performed on the Agilent 1260 Infinity Quaternary LC system equipped with a diode array detector (DAD) and coupled to TSKgel G3000SWXL, 5 µm, 7.8 mm × 30 cm column maintained at 30 °C. The mobile phase used was 10 mM histidine, 50 mM Arginine, 100 mM sodium sulfate, pH 6.0. The flow rate used was 1 mL/min. The eluted main and High-Molecular Weight (HMW) peaks were detected at 280 nm.

#### 2.2.4. Enzyme-Linked Immunosorbent Assay (ELISA)

A sandwich ELISA was performed using 96-well Maxisorp plates coated over-night at 2–8 °C with 1 μg/mL of an anti-idiotypic antibody that specifically recognizes 3BNC117-LS (anti-ID monoclonal antibody) or 1 μg/mL of an anti-idiotypic antibody that specifically recognizes 10-1074-LS (anti-ID monoclonal antibody). After washing, plates were blocked with 200 μL Protein free blocking solution at 25 °C for 2 h at 200 RPM. Co-formulated antibody samples, quality controls and reference standards were added and incubated at room temperature. Subsequently, the plate was washed and 100 μL of 1:10,000 diluted peroxidase-conjugated AffiniPure F (ab’) 2 Fragment Goat anti-Human IgG Fcγ Fragment specific (Jackson Immuno Research, West Grove, PA, USA) was added. The plate was incubated at room temperature for 60 ± 10 min at 200 RPM. The plate was washed, and the wells were incubated with 100 μL of SureBlue TMB substrate (Fisher Scientific, Somerset, NJ, USA) to develop the chromogenic signal (10 min at room temperature at 200 RPM). The reaction was stopped with the addition of 100 μL of 1% hydrochloric acid. The absorbance was measured at 450 nm using the Molecular Devices plate reader fitted with Softmax Pro software (Molecular Devices LLC, Sunnyvale, CA, USA). Titration curves for the reference standard and each test sample were created using 4-parameter logistic curve fitting to calculate EC50 values using GraphPad Prism software (version 7).

#### 2.2.5. Virus Neutralization Assays

The virus neutralization assay was evaluated using a luciferase-based assay in TZM-bl cells, as previously described [30,31]. Briefly, antibody samples were tested using a starting concentration of 25 μg/mL with 5-fold serial dilutions against the panel of HIV-1 Env pseudoviruses. The selected panel of HIV-1 Env pseudoviruses were either 3BNC117 sensitive/10-1074 resistant (*n* = 10) or 3BNC117 resistant/10-1074 sensitive (*n* = 10). The IC50 and IC80 titers were calculated as the mAb concentration that yielded a 50% or 80% reduction in relative luminescence units (RLU), respectively, compared to the virus control wells after the subtraction of cell control RLUs. All assays were performed in a laboratory compliant with Good Clinical Laboratory Practice (GCLP) procedures.

#### 2.2.6. FlowCAM^®^ Imaging 

FlowCAM^®^ is an imaging particle analysis system that we used for imaging and analyzing particles, in the subvisible range, using flow microscopy. The FlowCAM^®^ instrument (Fluid Imaging Technologies, Scarborough, ME, USA) was focused with 10 µm polystyrene beads at 3000/mL National Institute of Standards and Technology (NIST) standard. The samples were diluted by 4-fold by taking 200 µL of the sample in the corresponding buffer to a total volume of 0.8 mL, samples were analyzed at 0.08 mL/min through a 100 µm × 2 mm flow cell, and images of the particles were taken with a 10× optics system. Flash duration was set to 35.50 ms, and Camera Gain was set to 0. Visual-Spreadsheet software version 3.4.8 (DKSH Japan K.K., Tokyo, Japan) was used for data analysis.

#### 2.2.7. Osmolality

Osmolality, a measurement of the total number of solutes in a liquid solution expressed in osmoles of solute particles per kilogram of solvent (mOsm/Kg), was measured in the antibody formulations using the industry-preferred freezing point depression method. The osmolality measurements were made using a Model 3340 single-sample freezing-point micro-osmometer (Advanced Instruments, Norwood, MA, USA), equipped with a 20 µL Ease Eject™ Sampler (Parts No. 3M0825 and 3M0828). The unit of measurement used was milliosmoles of the solute per 1 kg of pure solvent, expressed as mOsm/kg. The instrument was calibrated with 50 mOsm/kg (3MA005) and 850 mOsm/kg (3MA085) calibration standards and verified with a 290 mOsm/kg Clinitrol^®^ Reference Solution (3MA029) prior to each analysis. 

#### 2.2.8. Dynamic Light Scattering (DLS)

Dynamic Light Scattering (DLS), which uses time-dependent fluctuations in the intensity of the scattered light to determine the effective size of a particle in nm range, was used to measure the particle size distribution in the antibody formulations. Dynamic light scattering was carried out at 25 °C, with a Malvern Zetasizer Nano Series instrument using a 633 nm/100 mW laser and a 90° detection angle. Particle size distribution (hydrodynamic diameter) by % intensity and % volume was determined along with the polydispersity index (PDI). 

## 3. Results

Parental anti-HIV antibodies, 10-1074 and 3BNC117, were individually formulated at 20 mg/mL for IV administration [15,17]. Based on initial pK data from phase 1 clinical studies, the parental antibodies were LS modified (as described in the Materials section) to extend serum half-life. Thereafter, as a first step towards formulation to aid subcutaneous administration, the antibodies were concentrated 7.5-fold in a new formulation buffer with optimal viscosity to enable drug injection volumes of 2 mL. This resulted in 3BNC117-LS and 10-1074-LS as individually formulated bnAbs, at 150 mg/mL in respective buffers (described in the Materials and Methods section), for subsequent co-formulation studies. 

These high-concentration individually formulated antibodies were extensively characterized using a wide range of analytical methods i.e., ELISA, SE-HPLC, RP-HPLC, IEX-HPLC, capillary isoelectric focusing (cIEF), CE-SDS (reduced and non-reduced), Sialic Acid analysis, intrinsic Tryptophan fluorescence spectroscopy, Isoquant analysis, dynamic light scattering (DLS), far and near UV circular dichroism (CD), differential scanning calorimetry (DSC), second derivative UV spectroscopy, N-terminal amino acid sequencing, HILIC based glycan profiling, liquid chromatography coupled with mass spectrometry (LC/MS), and peptide mapping by liquid chromatography coupled with tandem mass spectrometry (LC-MS/MS) (data not shown). In addition, both high-concentration bNAbs, in their respective formulation conditions, were found to be stable at 2–8 °C for ≥24 months (data not shown). 

To address the need to co-formulate both the antibodies as a single drug product at a combined final concentration of 150 mg/mL for subcutaneous administration in clinical studies, a 1:1 mixture of both bNAbs (each at 75 mg/mL) was formulated in 5 mM Histidine, 250 mM Trehalose, 10 mM Methionine, 5 mM Sodium Acetate, 0.05% Polysorbate 20, pH 5.5 buffer and characterized using a series of methods to test for (positive) identity, purity, product qualities, and functionality.

### 3.1. Chromatographic Separation of Co-Formulated Monoclonal Antibodies 

During analytical development, the aim was to select appropriate and optimal chromatographic techniques that could separate the two antibodies in their current co-formulation. Reverse phase high-performance liquid chromatography (RP-HPLC), ion exchange liquid chromatography (IEX), and size exclusion chromatography (SEC) were evaluated and found to achieve this separation goal. The separation efficiencies of each of these methods were challenged by the fact that both antibodies, 3BNC-117-LS and 10-1074-LS, are of the IgG1 subclass with similar molecular size and three-dimensional structure, and therefore significant method development and optimization of the chromatographic methods were necessary to achieve the desired separation goals. 

### 3.2. Reverse Phase High-Performance Liquid Chromatography (RP-HPLC)

Reverse phase liquid chromatography (RP-HPLC), due to the denaturing effect of the low pH and high organic solvent mobile phase, was expected to separate and quantify the two bNAbs, in the co-formulated milieu, based on their differences in the relative hydrophobicity. Initial assessment was performed on the Agilent 1260 Infinity quaternary LC coupled to a DAD detector using two columns: AdvanceBio RP-mAb Diphenyl, 2.1 × 100 mm, 3.5 µm (Agilent) and Accucore 150-C4 2.6 μm, 100 × 2.1 mm (Thermo). These columns represent two different stationary phase chemistries, diphenyl offers alternative selectivity and Accucore C-4 wide pore (150 Å) offers lower hydrophobic retention. A generic method development strategy was followed using 0.1% TFA in acetonitrile as mobile phase B, 0.5 mL/minute flow rate. Since, the individually formulated antibodies did elute at 30–40% acetonitrile at elevated temperature (60 °C column temperature) (data not shown), the initial gradient conditions for developing the method was set at 34% to 41% mobile phase B for 7 min. As part of the method optimization, different chromatographic conditions were tested to increase peak resolution: increasing temperature (45, 60, 70, and 80 °C), different mobile phases or organic modifiers in acetonitrile (methanol as mobile phase B or methanol/ IPA (5% *v*/*v*) as organic modifier in acetonitrile mobile phase), and different gradient conditions (34% B to 38% B for 16 min and 35% B to 39% B for 16 min). Finally, a method with 34% to 41% mobile phase B, and a column temperature of 60 °C, was selected that resulted in two separate peaks, corresponding to each monoclonal antibody (Figure 1A). This RP-HPLC method was then tested for linearity, precision, accuracy, and specificity parameters. Linearity was evaluated for the total peak area of 10-1074-LS and 3BNC117-LS in a co-formulated sample by calculation of a regression line using the least squares method. (Figure 1B). Linearity for the 3BNC117-LS specific area (Figure 1C) and 10-1074-LS specific area (Figure 1D) were also calculated; the *R*^2^ obtained was 1 for both analyses. Precision, for intra and inter day variability, was assessed by testing the repeatability of the target concentration of 2.5 µg (for each antibody) six times. The intra-assay precision for the total peak area ranged from 0.1% to 0.3% and the inter-assay precision was 0.3% for 3 experiments on separate days (days 1, 2, and 3) (Appendix A). The intra- and inter-precisions for the individual peak areas, 10-1074-LS peak area and 3BNC117-LS peak area, were also similar. The intra- and inter-precisions for the 10-1074-LS peak area were ≤0.4% and ≤0.3%, respectively (Appendix A). The intra- and inter-precisions for the 3BNC117-LS peak area were ≤0.2% and ≤0.2%, respectively (Appendix A). Accuracy was tested by percentage recoveries of the mean of three determinations of six different concentrations (1 to 8 µg column load). Based on the percent recoveries, we concluded that the RP-HPLC method accuracy was within a variation of ≤2% relative standard deviation (RSD) (Appendix A). These results indicate that this RP-HPLC method is suitable and, after appropriate method validation, can be used for future testing of the 3BNC117-LS + 10-1074-LS co-formulated drug product.

### 3.3. Ion Exchange High-Performance Liquid Chromatography (IEX-HPLC)

To allow the characterization of charge heterogeneity and high-resolution separation of each antibody (in the co-formulated sample), it was expected that the ion exchange (IEX) chromatography can separate the two bNAbs based on their charge differences. IEX chromatography is a non-denaturing technique and among the different IEX modes, since cation-exchange chromatography (CEX) is the preferred approach for characterizing antibody charge variants [32,33], it was chosen. The CEX method was developed on the Agilent 1260 Infinity quaternary LC system equipped with a solvent delivery pump, an autosampler, and a diode array detector (DAD). A ProPac WCX-10, 250 × 4 mm column (Thermo Scientific, Sunnyvale, CA, USA) was used for the method development. A “classical” salt gradient separation (50% to 100% B in 35 min) was performed using mobile phase A, composed of 20 mM Acetate buffer, pH 5.2, and mobile phase B, composed of 20 mM Acetate buffer containing 300 mM sodium chloride, pH 5.2. The flow rate was set to 0.7 mL/min and column temperature was maintained at 30 °C. Peak detection was carried out at 280 nm and after integration of peaks, the percentage peak areas of each peak (as well as charge variants i.e., main, acidic, basic peak) corresponding to each monoclonal antibody were calculated (Figure 2A). This weak cation exchange (CEX) chromatography method was used to perform qualitative and quantitative analysis of the charge variants for each of the separated antibodies. The optimized CEX-HPLC method was further tested for linearity, precision, accuracy, and specificity parameters. Linearity was evaluated for main peaks, pre-main peaks, post main peaks, and total peak areas of 10-1074-LS and 3BNC117-LS in a co-formulated sample by calculation of a regression line using the least squares method (Figure 2B–D). Precision, for intra- and inter-assay variability, of the total peak area was assessed by testing the repeatability of the target concentration of 100 µg (total) six times. The charge variants for both 10-1074-LS and 3BNC117-LS were within 2.2% for the total peak area with intra-assay precision within 1.1–2.2% and inter-assay precision within <2.2% (Appendix A). The variability of the 10-1074-LS specific peak was similar, with intra-assay precision ≤ 2.1% and inter-assay precision ≤ 1.8% (Appendix A). The variability of the 3BNC117-LS specific peak was slightly higher, although similar, with intra-assay precision ≤ 2.6% and inter-assay precision ≤ 2.7% (Appendix A). The accuracy was tested by percentage recoveries of the mean of three determinations of six different concentrations (50 to 300 µg column load); the method accuracy was observed to be ≤2% relative standard deviation (RSD) (Appendix A). Based on these results, this CEX-HPLC method is suitable and after appropriate method validation can be used for future testing of the 3BNC117-LS + 10-1074-LS co-formulated drug product.

### 3.4. Size-Exclusion High-Performance Liquid Chromatography (SE-HPLC)

To separate the two antibodies (in the co-formulated sample) based on their molecular size and achieve separation through differential exclusion, we used Size Exclusion HPLC (SE-HPLC). SE-HPLC is widely used for determining the antibody purity, through the determination of percent monomer (and assessments of % HMW, High Molecular Weight, and % LMW, Low Molecular Weight, species), and therefore we were conscious of the possible limitation of the method to fully resolve the two similarly sized monoclonal antibodies in the co-formulated sample. When we evaluated two mobile phases (100 mM sodium acetate (pH 6.0) and 100 mM sodium sulfate (pH 6.0)) for the resolution of the antibodies in the co-formulated sample, we found that both phases resulted in one broad peak with no resolution of the two antibodies (data not shown). However, when we changed the mobile phase to 10 mM histidine, 50 mM arginine, 100 mM sodium sulfate, pH 6.0, a slight separation between the two peaks was observed (Figure 3A). When the salt (sodium sulfate) concentration was gradually increased from 100 to 550 mM sodium sulfate, to probe the effect of increasing salt on the separation of the two peaks (10-1074-LS and 3BNC117-LS peaks), we observed separation with greater resolution, despite the broadening of the late eluting 10-1074-LS peak. From the outset, since the intent of the SE-HPLC was not to resolve the two mAbs but to detect the levels of HMW (and LMW) species in the co-formulated sample and quantify aggregate levels (at the time of product release and during long-term storage) to ensure a means for measurement of percent monomeric antibody in the co-formulated milieu, this SE-HPLC method was accepted to be appropriate for use and tested further for linearity, precision, accuracy, and specificity parameters. Linearity was verified in a range of co-formulated samples with a R^2^ of >0.99 (Figure 3B–D). The intra- and inter-precision at the target column load of 100 µg (total) for the main peak (monomer) area was within ≤0.2% (Appendix A). The intra- and inter-precision at the target column load of 100 µg (total) for the % HMW peak area was within ≤0.7% and ≤4.3% (Appendix A). However, the intra- and inter-precision at the target column load of 100 µg (total) for the % LMW peak area was higher, within ≤18.4% and ≤14% (Appendix A); this higher % RSD was due to lower signal levels (lower levels of LMW), closer or below limit of quantification (LOQ). Accuracy was tested by percentage recoveries of the mean of three determinations of six different concentrations precisely prepared (50 to 300 µg column load); the method accuracy was observed to be within 2% RSD (Appendix A). These results indicate that the SE-HPLC method is suitable and after appropriate method validation can be used for future testing of the 3BNC117-LS + 10-1074-LS co-formulated drug product.

### 3.5. Positive Identification of Individual Antibody in the Co-Formulation Sample Using Anti-ID (Idiotype) Based ELISA

Since wildtype gp120-based ELISA would not be successful in differentiating the binding and the identity of the two antibodies, when present in a co-formulated sample, we generated a 10-1074 idiotype-specific antibody and a 3BNC117 idiotype-specific antibody to serve as reagents in a new ELISA that would utilize each antibody’s identity based on their unique idiotype (ID). This format would provide a means for measuring the identity of an individual antibody in the co-formulated sample. After a series of optimization experiments, the anti-ID ELISA was successful to identify and differentiate both antibodies as well as detect their identity in the co-formulated sample (Figure 4). The anti-ID ELISA was tested for precision and accuracy (data not shown) and the overall variability, particularly inter-assay, was well within the 30–40% RSD, seen in bioassays (data not shown). These results indicate that the anti-ID based ELISA can be used for future testing of identity of the 3BNC117-LS + 10-1074-LS co-formulated drug product, after appropriate method validation.

### 3.6. Potency Testing of Individual Antibody in the Co-Formulation Sample Using a Virus Neutralization Assay

A traditional HIV-1 pseudovirus neutralization assay was used to evaluate the functional activity or potency of the individual antibodies in the co-formulated sample. To do so, two panels of pseudoviruses (total *n* = 20) were selected: one panel (*n* = 10) for 3BNC117 and another (*n* = 10) for 10-1074. To test for 3BNC117 potency/functional activity, the panel involved ten 3BNC117 sensitive/10-1074 resistant viruses; to test for 10-1074 potency/functional activity, the panel involved ten 3BNC117 resistant/10-1074 sensitive viruses. Viruses were selected for having low/medium to high sensitivity to a single antibody based on historical data (data not shown). MuLV (Murine Leukemia Virus), a non-relevant virus, was used as a negative control and was not neutralized by either of the two control antibodies (data not shown). In comparison to the individual antibody (3BNC117 or 10-1074), used as control, the two co-formulated antibodies were potent and demonstrated their specific neutralization activity in their respective panels (Table 1A–D). Out of the 10 viruses sensitive to 3BNC117, 0013095-2.11 is not highly sensitive to 3BNC117; therefore, not only is the IC80 >25 μg/mL, the IC80 of the co-formulated 3BNC117-LS and 10-1074-LS is also higher when compared to IC80s for other viruses for the co-formulated product. (Table 1A). Despite this one virus, which could be replaced by another virus in future, based on these results, the pseudovirus neutralization assay using the defined panel of viruses can be used for future testing of the 3BNC117-LS + 10-1074-LS co-formulated drug product. 

### 3.7. Stability Assessment of Co-Formulated Antibodies

To assess the stability of the co-formulated antibodies at high-concentration in the chosen formulation, we performed a 28 day short stability study that included both a real-time stability study in storage conditions (i.e., at 5 ± 3 °C) and a study of samples in accelerated (25 ± 2 °C/RH 60% ± 5%) and stressed (40 ± 2 °C/RH 75% ± 5%) conditions. RH here refers to Relative Humidity.

For the real-time stability study, we used 2.0 mL co-formulated samples in 3.0 mL Schott glass vials and incubated the samples at 5 ± 3 °C. For evaluation at accelerated and stressed conditions, we used similar sample volumes in 3.0 mL Schott vials and incubated them at 25 ± 2 °C/RH 60% ± 5% (accelerated conditions) and 40 ± 2 °C/RH 75% ± 5% (stressed conditions). Samples were analyzed at T = 0, before study start, and thereafter samples were collected on a weekly basis and analyzed for visual appearance, pH, total protein concentration by UV spectroscopy (280 nm), purity (by determining % monomer and HMW aggregates) by SE-HPLC, charge variants (i.e., relative levels of acidic and basic species) by CEX-HPLC, content of individual antibody by RP-HPLC, protein degradation by SDS-PAGE, sub-visible particles by *FlowCAM*^®^ instrument, viscosity by Viscosizer TD, and (hydrodynamic) particle size by DLS. At real-time storage conditions, the antibodies were stable in the co-formulated milieu for up to 4 weeks across all test parameters (Table 2). In addition, the antibodies were also stable in the accelerated conditions, 25 ± 2 °C/RH 60% ± 5%, for up to 4 weeks across all test parameters (Appendix A). Furthermore, the co-formulated antibodies were stable up to 4 weeks at stressed conditions, 40 ± 2 °C/RH 75% ± 5%, (Appendix A). 

In addition to the above analysis, a limited set of samples (T = 2 weeks and T = 4 weeks) from all 3 (real-time, accelerated, and stressed) conditions were tested for potency (functional activity) of the antibodies using pseudovirus neutralization assay. When compared to unincubated co-formulated samples (control) (Table 1A–D), all samples were found to neutralize the pseudoviruses with little to no change in IC50 and 1C80 values and hence found to be stable for up to 4 weeks (Appendix A). These results indicate not only the utility of the various assays in monitoring antibody stability in the co-formulated sample but also highlight the stability and suitability of the formulation in co-formulating the two antibodies.

This initial stability assessment and identification of appropriate analytical assays support the clinical development of these co-formulated drug products for future clinical studies. 

## 4. Discussion

After the initial evaluation of passive administration of first generation anti-HIV antibodies (4E10+2F5+2G12) [34], the identification of a large number of next-generation anti-HIV bNAbs with greater breadth and potency in the past decade has opened the possibility for antibody-based treatment and/or prevention of HIV-1 infection. Several bNAbs have recently progressed to clinical trials in humans: VRC01 [13,14]/VRC01LS, 10-1074 [15]/10-1074-LS [16], 3BNC117 [17]/3BNC117-LS [18], VRC07-523-LS [19], PGT121 [20,21], and PGDM1400 [21] or their combinations. These early (phase I) clinical studies, with safety, pharmacokinetics, and viral load re-bound or decay as endpoints, have primarily used antibodies formulated for IV infusion. However, to overcome the vast diversity of HIV-1 variants, it is becoming increasingly clear that combinations of (two or more) bNAbs targeting distinct epitopes on the viral envelope (Env) will likely be required [35]. To support the development of bNAb combinations as products for clinical studies, co-formulating two or more antibodies, targeting different Env epitopes, as a single drug product and using a subcutaneous (SC) route for administration, are under consideration in multiple clinical studies. To that end, not only the high-concentration formulation of two or more antibodies in a limited volume will be necessary, but methods to test their individual quality attributes (e.g., purity, charge variants, potency) of the individual antibody in the co-formulated milieu will be required [35]. 

In this study, two anti-HIV-1 antibodies, 3BNC117-LS and 10-1074-LS, were co-formulated in a 1:1 ratio to achieve a final concentration of 150 mg/mL in 5 mM Histidine, 250 mM Trehalose, 10 mM Methionine, 5 mM Sodium Acetate, 0.05% Polysorbate 20, pH 5.5 buffer. To support the high-concentration formulation and development of the two antibodies for subcutaneous administration, formulation optimizations and analytical test method development and optimizations were performed. Analytical characterization and separation of individual antibodies in the co-formulated sample was challenging due to the high degree of similarity in the physico-chemical properties of the two (3BNC117-LS, and 10-1074-LS) antibodies. Systematic analytical development was carried out, using several methodologies, to obtain separation of the two antibodies. Specifically, chromatographic methods were developed to resolve and assess the quality attributes of the individual antibody in the co-formulated drug product. RP-HPLC and CEX-HPLC methods resulted in baseline separation of the two antibodies (3BNC117-LS and 10-1074-LS) in the co-formulated sample, and the peak profiles compared well to the individually (high-concentration) formulated antibodies. The SE-HPLC method was used to assess the combined high molecular weight species of the two antibodies; the data showed partially separated peaks, corresponding to the two co-formulated antibodies, with no additional % HMW species at this stage. Further evaluation of all HPLC methods for specificity, purity, accuracy, precision, and repeatability confirms that the methods are suitable for future testing of the such co-formulated antibody-based drug product. 

In addition to the HPLC methods, an anti-ID ELISA was developed to test identity of the individual antibodies in the co-formulated drug product. In addition, the utilization of a separate and well-defined pseudovirus panel in a virus neutralization assay provided a functional assay platform to not only evaluate the potency/functionality of the individual antibodies but also an approach to test two (or more) antibodies via this functional assay.

In summary, through demonstration of the high-concentration co-formulation of two anti-HIV-1 antibodies and the development of separation-based testing methods, we present several analytical tools to test physico-chemical and functional attributes of co-formulated antibodies, which can contribute to the clinical development of these high-concentration antibodies. Finally, the little to no inter-molecular protein–protein interaction between the antibodies, even at ≥150 mg/mL, and their stability profile ensure the possibility of the development of such high-concentration antibodies as products for HIV prevention and/or treatment.

## Figures and Tables

**Figure 1 antibodies-09-00036-f001:**
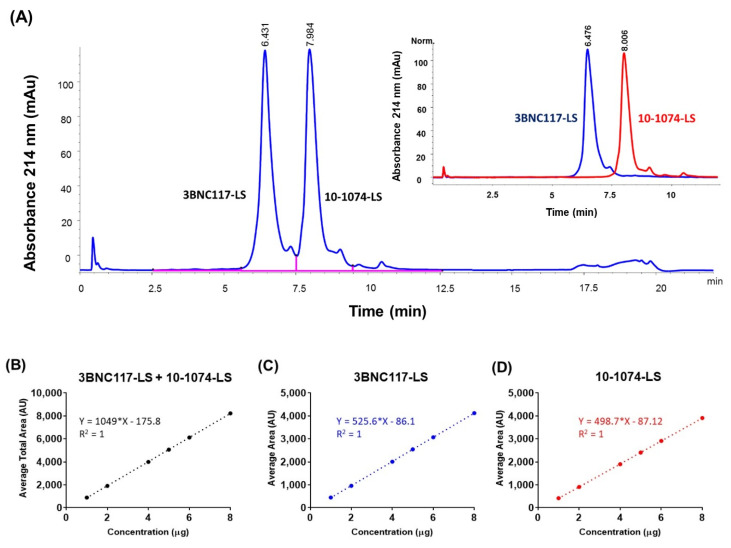
Reverse phase chromatogram with UV absorbance at 214 nm showing separation of two antibodies in the 1:1 co-formulated sample (total 150 mg/mL). (**A**) Peaks separated by reversed phase HPLC corresponding to the two antibodies are labeled. Inset shows the overlapped reverse phase chromatograms from the two-separate RP-HPLC run corresponding to the two antibodies, each at 150 mg/mL. The chromatography conditions were the same for the co-formulated and the individual antibody samples. Linearity analysis of concentration (in μg; x-axis) dependent increase in area under the curve (in Absorbance Units, AU; y-axis) for (**B**) total area (3BNC117-LS + 10-1074-LS), (**C**) 3BNC117-LS specific area, and (**D**) 10-1074-LS specific area.

**Figure 2 antibodies-09-00036-f002:**
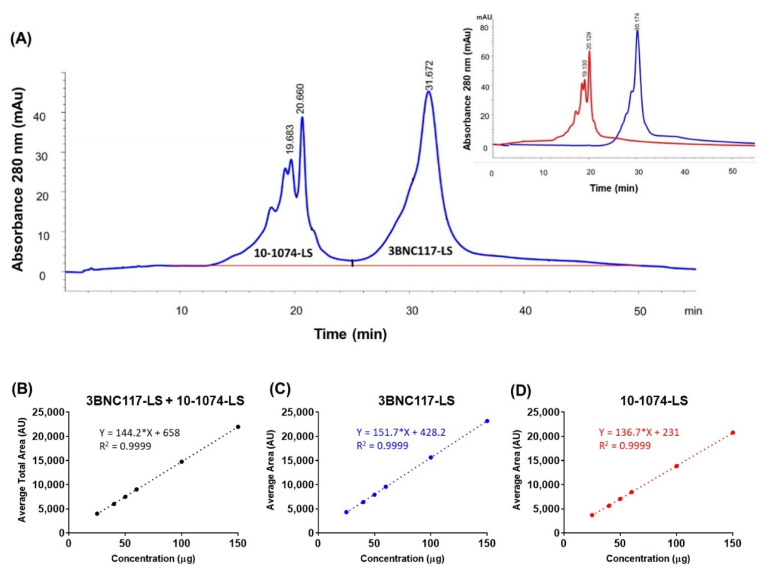
(**A**) Cation exchange chromatogram with UV absorbance at 280 nm showing separation of two antibodies in the 1:1 co-formulated sample (total 150 mg/mL). Peaks for each of the antibody are labeled. Inset shows the overlapped cation exchange chromatograms from the two separate chromatography runs corresponding to the two antibodies, each at 150 mg/mL. The chromatography conditions were the same for the co-formulated and the individual antibody samples. Linearity analysis of concentration (in μg; x-axis) dependent increase in area under the curve (in Absorbance Units, AU; y-axis) for (**B**) total area (3BNC117-LS + 10-1074-LS), (**C**) 3BNC117-LS specific area, and (**D**) 10-1074-LS specific area.

**Figure 3 antibodies-09-00036-f003:**
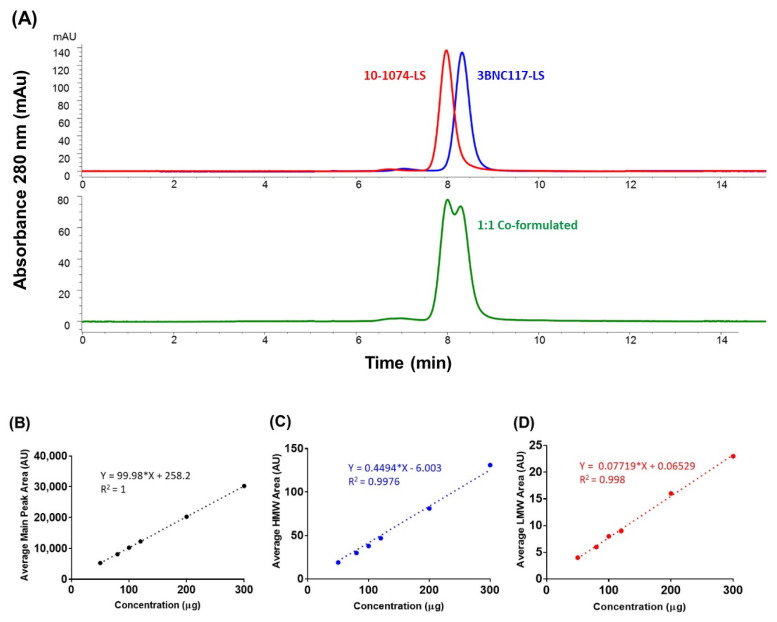
(**A**) Size exclusion chromatography profiles of 150 mg/mL for 10-1074-LS and 3BNC117-LS antibodies, individually formulated (top) and after co-formulation (1:1, 75 mg/mL each) (bottom) on the TSKgel column. Linearity analysis of concentration (in μg; x-axis) dependent increase in area under the curve (in Absorbance Units, AU; y-axis) for (**B**) average main peak (monomer) area (3BNC117-LS + 10-1074-LS monomer), (**C**) average High-Molecular Weight (HMW) peak area, and (**D**) average Low Molecular Weight peak area.

**Figure 4 antibodies-09-00036-f004:**
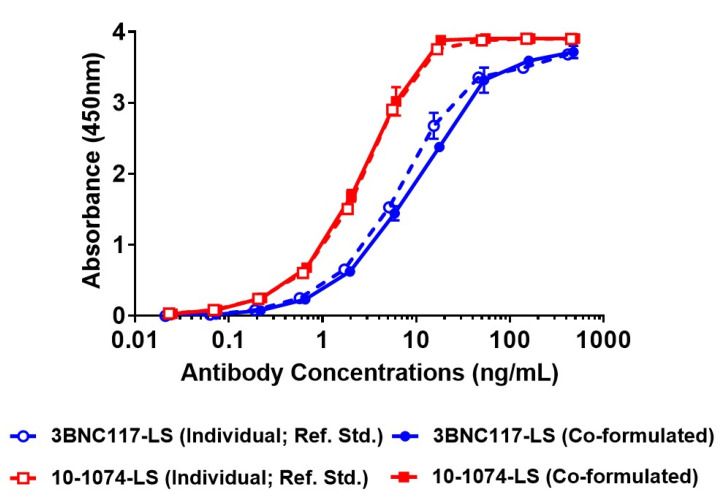
Test of identity of individual antibodies (3BNC117-LS and 10-1074-LS) in co-formulated sample (of 150 mg/mL total concentration) using anti-idiotypic (anti-ID) antibodies. Individual antibodies, 3BNC117-LS and 10-1074-LS at 150 mg/mL, were used as a reference standard. Open circle, dotted line—3BNC117-LS (150 mg/mL, reference standard), filled circle, filled line—3BNC117-LS (at 75 mg/mL in co-formulated sample), open square, dotted line—10-1074-LS (150 mg/mL, reference standard), filled square, filled line—10-1074-LS (at 75 mg/mL in co-formulated sample).

**Table 1 antibodies-09-00036-t001:** Neutralization activity of co-formulated 3BNC117-LS and 10-1074-LS (total 150 mg/mL) using 2 panels of pseudoviruses in TZM-bl cells. One panel (**A**,**B**) is used to test potency/functional activity of 3BNC117 (3BNC117 sensitive/10-1074 resistant viruses (*n* = 10)) and the other panel (**C**,**D**) is used to test potency/functional activity of 10-1074 (10-1074 sensitive viruses/3BNC117 resistant (*n* = 10)). The pseudovirus strains are indicated at the top of the table and IC50 and IC80 values (in μg/mL) for each of the samples, against those viruses, are reported. Individual antibodies, 3BNC117 and 10-1074, are used as controls for each panel. LS—Leucine-Serine substitution.

(**A**)										
**Samples**	**ZM249M.PL1**		**Q461.e2**		**0013095-2.11**		**62357.14.D3.4589**		**ZM53M.PB12**	
3BNC117.LS + 10-1074.LS DP	IC50	IC80	IC50	IC80	IC50	IC80	IC50	IC80	IC50	IC80
	0.042	0.15	0.042	0.153	1.161	15.162	0.043	0.15	0.153	0.568
3BNC117.LS (control)	0.037	0.13	0.039	0.143	1.396	>25	0.036	0.17	0.214	0.796
10-1074.LS (control)	>25	>25	>25	>25	>25	>25	>25	>25	>25	>25
(**B**)										
**Samples**	**C2101.c01**		**C4118.c09**		**THRO4156.18**		**415.v1.c1**		**CNE5**	
3BNC117.LS + 10-1074.LS DP	IC50	IC80	IC50	IC80	IC50	IC80	IC50	IC80	IC50	IC80
	0.029	0.135	0.034	0.162	1.498	8.209	0.05	0.115	0.193	0.898
3BNC117.LS (control)	0.044	0.15	0.051	0.183	1.939	9.815	0.048	0.142	0.193	0.911
10-1074.LS (control)	>25	>25	>25	>25	>25	>25	>25	>25	>25	>25
(**C**)										
**Samples**	**1394C9_G1 (Rev-**)		**ZM247v1 (Rev-**)		**Du422.1**		**6631.v3.c10**		**377.v4.c9**	
3BNC117.LS + 10-1074.LS DP	IC50	IC80	IC50	IC80	IC50	IC80	IC50	IC80	IC50	IC80
	0.02	0.077	0.02	0.099	0.032	0.114	0.157	0.796	0.418	1.397
3BNC117.LS (control)	>25	>25	>25	>25	>25	>25	>25	>25	>25	>25
10-1074.LS (control)	0.033	0.119	0.036	0.162	0.045	0.161	0.189	0.968	0.433	1.515
(**D**)										
**Samples**	**20915593**		**T278-50**		**21197826-V1**		**Du151.2**		**19715820_A10_H2**	
3BNC117-LS + 10-1074-LS DP	IC50	IC80	IC50	IC80	IC50	IC80	IC50	IC80	IC50	IC80
	1.525	5.872	1.047	11.952	0.678	2.269	0.004	0.013	0.056	0.204
3BNC117-LS (Control)	>25	>25	>25	>25	>25	>25	>25	>25	>25	>25
10-1074-LS (control)	2.02	5.817	2.174	15.13	0.613	2.188	0.005	0.015	0.074	0.253

**Table 2 antibodies-09-00036-t002:** Summary of 28 day stability testing results of co-formulated antibodies, 3BNC117-LS and 10-1074-LS (total 150 mg/mL), evaluated at 0, 1, 2, 3, and 4 weeks, after incubation at storage conditions of 5 ± 3 °C. HMW = High Molecular Weight; d.nm = Diameter in nm; PDI = Polydispersity Index; P/mL = Particles/mL.

Test Attributes		Weeks				
		0	1	2	3	4
pH		5.65	5.6	5.62	5.60	5.59
A280 (mg/mL)		142	137	142	139	149
Viscosity (cP)		10.70	11.08	12.09	11.16	12.89
Osmolality (mOsm/Kg)		345	336	333	336	337
SE-HPLC	HMW (%)	2.98	3.11	3.14	3.58	3.52
	Main Peak (%)	96.90	96.81	96.84	96.41	96.46
CEX-HPLC 3BNC117-LS	Main Peak (%)	48.78	49.08	49.15	49.27	49.71
	Pre-Main Peaks (%)	47.68	46.40	46.04	44.74	44.97
	Post-Main Peaks (%)	3.54	4.52	4.81	5.99	5.32
CEX-HPLC 10-1074-LS	Main Peak (%)	32.68	35.04	35.22	35.99	37.72
	Pre-Main Peaks (%)	61.64	59.70	59.55	59.07	57.45
	Post-Main Peaks (%)	5.68	5.26	5.23	4.94	4.83
RP-HPLC	3BNC117-LS (mg/mL)	68.05	65.88	70.00	68.70	71.48
	10-1074-LS (mg/mL)	76.70	75.75	80.54	78.83	82.09
DLS	Z-Average (d.nm)	10.25	10.44	10.26	10.23	10.30
	PDI	0.18	0.21	0.19	0.18	0.19
FlowCAM	2–10 µm (P/mL)	191	101	253	126	475
	10–25 µm (P/mL)	31	23	46	36	107
	25–50 µm (P/mL)	15	16	8	9	8

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
