# Peer review of "Characterization of Co-Formulated High-Concentration Broadly Neutralizing Anti-HIV-1 Monoclonal Antibodies for Subcutaneous Administration"

_2073-4468, 2020, doi:10.3390/antib9030036_

Round 1

Reviewer 1 Report

Sharma et al., analyzed the quality control data of a formulation in which two antibodies 3BNC117-LS and 22 10-1074-LS were mixed. This co-formulated antibodies are important for the future development of HIV treatment, and although the significance of the research is recognized, it is considered to have little scientific merit. If PK data of subcutaneous administration of this formulation are available, the scientific merit of this manuscript may significantly increase.

Table 1 shows data comparing neutralizing activity of co-formulated form with each antibody. Most of the data are consistent with the neutralizing activity of each antibody, but only in the case of samples 0013095-2.11 the data looks different. In other words, viruses that cannot be neutralized with the respective antibodies can be neutralized with the co-formulated form. Some explanation have to be done in the text.

Minor point: The space between the rows in Table 1 is not constant, and there are some cases where the position of the data is inappropriate.

Author Response

Dear Reviewer,

We thank you for your kind review of research manuscript entitled “Characterization of co-formulated high-concentration broadly neutralizing anti-HIV-1 monoclonal antibodies for subcutaneous administration” by Vaneet Sharma et al. We have responded to all your comments (highlighted in red in the attached word file), addressed them and have revised the manuscript accordingly.

We now submit the revised manuscript for your final review. Thank you for your consideration.

best,
Antu K. Dey, DPhil.
Executive Director, Product Development and Manufacturing,
International AIDS Vaccine Initiative,
125 Broad St, New York, NY 10004, USA.

Reviewer 2 Report

This article by Sharma et al describes the characterization of the physico-chemical and functional properties of two anti-HIV Envelope (Env) specific broadly neutralizing antibodies (bNAbs, 3BNC117-LS and 10-1074-LS) co-formulated together and how these compare to the properties of each of these bNAbs at equivalent concentrations formulated separately.

The rationale for this effort derives from the increasing interest in using bNAbs therapeutically or as prevention for HIV infection. Several bNAbs have progressed to clinical trials. Because of extensive variation in HIV and particularly in HIV Env sequences present between- and within-HIV+ individuals, there is an advantage to using combinations of 2 or more bNAbs targeting different Env epitopes rather than one. To date, most of these bNAbs have been administered intravenously. However, there is interest in delivering these Abs subcutaneously. For such an application, each bNAb would have to be formulated at a high concentration to be delivered this way in a small volume.

In this manuscript, the authors report on the optimization strategies used for developing tests to be performed on the 2 bNAbs co-formulated together at a high concentration and compared the performance of these Abs to that of the two individual Abs that make up the co-formulation. In order to assess the individual quality attributes of the Abs co-formulated together both of which have similar physico-chemical properties, the tests used to assess quality had to be successfully optimized.

The authors make a convincing case for their ability to separate the Abs using reverse phase HPLC and ion exchange liquid chromatography. Size exclusion chromatography was less successful at resolving the 2 bNAbs but was deemed to be useful for detecting low and high molecular weight species and aggregates, which would be useful for testing these co-formulated drug products before release.

They developed an ELISA assays using anti-idiotypic Abs to the Ag binding site of 3BNC117-LS and 10-1074-LS showing that single Abs performed similarly to the co-formulated version of the Abs. Neutralizing Ab tests were performed on a panel of pseudoviruses half of which were resistant to 3BNC117-LS and susceptible to 10-1074-LS and the other half susceptible to 3BNC117-LS and resistant to 10-1074-LS. They also performed stability testing under 3 conditions over a 4 week period to assess changes in pH, protein concentration, aggregate formation viscosity among other assessments.

Overall, the manuscript describes a thorough series of testing aimed at assessing the quality of co-formulated Abs that should be transferrable to the assessment of other coformulations of bNAbs. I only have minor comment to make on this manuscript.

Minor comments

The word “The” is missing on lines 17, 147, 249, 259, 280, 292

The word “a” is missing on lines 259, 260, 357, 365, 384, 390

The sentence on lines 69-71 should be rewritten to correct English usage. 

Please explain what FloCAM Imaging, Osmolality and Dynamic light scattering (DLS) are, what they measure and how they are being used here.       

Line 177, it is mentioned that the parental Ab were LS modified as described in the materials section. This modification is not described in this section.

Line 271 “peaks area” should be “peak areas”.

Fix text on line 327/328.

Line 365 “of the two antibodies” should be “of the two controls” or “of the two control antibodies”

Line 377 control should be “controls”

Line 386 define RH

Line 394 insert “using a”

Author Response

(The authors gave the same response as above.)
